# Effect of Drying Methods on the Morphological and Functional Properties of Cellulose Ester Films

**DOI:** 10.3390/polym17223026

**Published:** 2025-11-14

**Authors:** Tanuj Kattamanchi, Heikko Kallakas, Elvira Tarasova, Percy Festus Alao, Tiit Kaljuvee, Arvo Mere, Atanas Katerski, Rünno Lõhmus, Andres Krumme, Jaan Kers

**Affiliations:** 1Laboratory of Wood Technology, Department of Materials and Environmental Technology, School of Engineering, Tallinn University of Technology, Ehitajate tee 5, 19086 Tallinn, Estonia; tanuj.kattamanchi@taltech.ee (T.K.); heikko.kallakas@taltech.ee (H.K.); percy.alao@taltech.ee (P.F.A.); 2Laboratory of Biopolymer Technology, Department of Materials and Environmental Technology, School of Engineering, Tallinn University of Technology, Ehitajate tee 5, 19086 Tallinn, Estonia; elvira.tarasova@taltech.ee (E.T.); andres.krumme@taltech.ee (A.K.); 3Laboratory of Inorganic Materials, Tallinn University of Technology, Ehitajate tee 5, 19086 Tallinn, Estonia; tiit.kaljuvee@taltech.ee; 4Laboratory for Thin Film Energy Materials, Department of Materials and Environmental Technology, Tallinn University of Technology, Ehitajate tee 5, 19086 Tallinn, Estonia; arvo.mere@taltech.ee (A.M.); atanas.katerski@taltech.ee (A.K.); 5Laboratory of Nanostructures Physics, Institute of Physics, University of Tartu, W. Ostwaldi 1, 50411 Tartu, Estonia; rynno.lohmus@ut.ee

**Keywords:** solvent casting, vacuum drying, [mTBNH][OAc], cellulose esters, polymer films, EIPS

## Abstract

This study presents the synthesis and characterisation of cellulose long chain fatty acid ester films using a novel distillable ionic liquid (IL), 5-methyl-1,5,7-triaza-bicyclo-[4.3.0] non-6-enium acetate [mTBNH][OAc] in combination with DMSO as a cosolvent. The cellulose esters cellulose diacetate (CDA), cellulose laurate (CL), and cellulose palmitate (CP) were fabricated through an evaporation-induced phase separation method (EIPS) and dried under two conditions: conventional oven drying (RO) and vacuum oven drying (VO). The influence of drying conditions on the structural, thermal, and surface properties of the films was evaluated using XRD, TGA, SEM, AFM, and contact angle measurement techniques. XRD confirmed an amorphous structure in all films, with no significant effect on the drying conditions. TGA revealed consistent thermal degradation profiles across all samples, with ester group decomposition accruing between 140 and 250 °C and main cellulose backbone degradation near 350 °C. The SEM cross-section showed a uniform film, devoid of cavities and layered structures. AFM analysis demonstrated that VO-dried films had smoother surfaces compared to RO-dried films, correlating with increased contact angles and enhanced hydrophobicity. A strong inverse relationship between surface roughness and hydrophobicity was observed, particularly in VO-dried samples, although this was not statistically significant due to data variability. Overall, the drying method had minimal impact on the internal structure and thermal stability; it significantly influenced surface morphology and wettability.

## 1. Introduction

The growing interest in bioplastics stems from their environmental benefits and potential to reduce reliance on fossil-based materials. Cellulose, the most abundant biopolymer in nature, has gained attention due to its renewable source and inherent properties [1]. Despite its advantageous attributes, cellulose is inherently non-thermoplastic, limiting its application in the plastics industry. However, through chemical modification, particularly esterification, cellulose can be transformed into thermoplastic materials with enhanced processability and functional properties [2]. The esterification of cellulose with long-chain fatty acids has emerged as a widely used method to improve its thermoplastic behaviour, producing cellulose esters (CE) like cellulose acetate (CA) [3]. These derivatives possess better moldability, making them suitable for a variety of applications in bioplastics and beyond [4].

The crystallinity of cellulose plays a key role in its reactivity. The crystalline structure impedes the ability of cellulose to interact with solvents and reagents, making its modification challenging [5]. While cellulose is hydrophilic due to its hydroxyl groups, the amorphous regions within the polymer are more susceptible to solvent penetration, allowing for chemical modification and improving the material’s overall performance [6]. The production of cellulose esters, however, is often limited by the high crystallinity of the polymer, which affects its reactivity and can lead to degradation during the esterification process. This challenge is typically addressed through two main methods: homogeneous and heterogeneous esterification. The traditional heterogeneous method, which employs sulfuric acid as a catalyst, is labour-intensive and prone to degradation due to the low reactivity of crystalline cellulose. In contrast, homogeneous esterification offers a more controlled process, resulting in cellulose esters with higher degrees of substitution and more favourable physical properties [7].

The synthesised cellulose esters are typically processed into films, which are essential for a variety of applications, such as packaging materials and biodegradable films. One common method for film preparation is solvent casting, where a solution of cellulose ester is applied to a substrate, and the solvent is evaporated under controlled conditions. The solution casting technique is a simple yet reliable approach for film fabrication [8], where the polymer solution is uniformly dispensed onto a substrate, followed by solvent evaporation under controlled or partially controlled conditions to yield films. Rapid solvent evaporation during film formation has garnered increasing attention due to its operational simplicity and cost effectiveness [9,10]. In such methods, solutions of hydrophobic polymers are subjected to accelerated solvent evaporation, resulting in the formation of surface morphologies. The resulting topological features are closely associated with enhanced water repellency [11,12]. The impact of experimental drying parameters on film properties (temperature, pressure, etc.) such as wettability and topography, has been explored minimally in the literature. Studies were reported that drying conditions have shown to significantly influence the morphology, optical appearance, and functional properties of cellulose-based films. Lyytikäinen et al. (2021) demonstrated that drying temperature affects wettability and barrier performance, while substrate roughness can induce air entrapment, altering the surface morphology [13]. Harini and Sukumar (2019) observed that vacuum drying yields transparent cellulose acetate films, whereas ambient drying results in opacity, highlighting the critical role of the drying environment [14]. These findings emphasize the importance of controlling drying parameters to optimise film properties. Environmental factors such as temperature, humidity, and pressure during drying can significantly influence the properties of the resulting films [13]. For instance, drying temperature has been shown to affect the film’s wetting behaviour and barrier properties by altering the film’s microstructure. The literature indicates that the drying technique can determine whether the film becomes transparent or opaque, likely due to differences in the evaporation rates of the solvent and the retention of solvent within the film structure [14].

Despite the known influence of drying conditions on the properties of cellulose ester films, there remains a lack of literature understanding how these variables affect the film’s physical, chemical, and thermal properties on long chain cellulose esters. This study aims to fill this gap by systematically investigating the effects of drying conditions on the physical properties of long-chain cellulose ester films.

This study tests the hypothesis that drying conditions significantly influence the physical properties of long chain cellulose ester films. By systematically comparing vacuum oven (VO) and forced air circulation (RO) drying at elevated temperatures, the research investigates how solvent evaporation dynamics affect film morphology, thermal stability, and surface characteristics. It is poised that the drying environment governs film formation mechanisms, thereby impacting the uniformity and performance of the resulting materials. Comparison with previously reported room temperature air-dried films provides further insight into how the drying parameters affect the morphology, structure, thermal, and wetting properties [15].

The outcomes of this study will provide valuable information about the influence of drying techniques on the properties of cellulose ester films. By analysing the thermal, chemical, and morphological characteristics, this research will contribute to optimising the processing conditions for cellulose-based bioplastics, paving the way for more efficient and widespread application. Understanding how drying conditions impact the properties of cellulose ester films is crucial for enhancing their performance in practical applications, such as biodegradable packaging, agricultural films, and other environmentally friendly products.

## 2. Materials and Methods

### 2.1. Materials

The fibrous cellulose was purchased from Carl Roth GMBH (Karlsruhe, Germany). Vinyl laurate and vinyl palmitate with purity >98% were purchased from Tokyo Chemical Industry Co. (Tokyo, Japan). Novel ionic liquid 5-Methyl-1,5,7-triaza-bicyclo-[4.3.0]non-6-enium acetate [mTBNH][OAC] was specifically synthesised for this research by Liuotin Group Oy (Porvoo, Finland). The melting point of IL is 15 °C; the flash point is more than 220 °C. DMSO with a purity of 99.9% was purchased from Fisher Chemical (Pittsburgh, PA, USA). Cellulose acetate (CDA) powder (Mn = 30,000 g/mol, acetyl content 39.8%) and pyridine (purity > 99%) were purchased from Sigma-Aldrich (St. Louis, MO, USA).

#### 2.1.1. Cellulose Transesterification

Fibrous cellulose (CAS No. 9004-34-6) was procured from Carl Roth GmbH & Co. KG (Karlsruhe, Germany) with the average fibre length being between 0.02 and 0.1 mm, a density of 1.5 g/cm^3^, and a pH range of 5–7. Following the ASTM D1795-13 standard [16], the average molar mass (MM) of cellulose was determined using the intrinsic viscosity [η] measurement in cupriethylenediamine hydroxide (CuEn) at 25 °C [17]. Subsequently, we used the Mark-Houwink equation with the parameters K = 1.01 × 10^−4^ dL/g and a = 0.9 [18], yielding a molar mass value 163,000 g/mol. The transesterification reaction of this cellulose was carried out with vinyl esters, using ionic liquid (IL), i.e., 5-methyl-1,5,7-triaza-bicyclo-[4.3.0]non-6-enium acetate, [mTBNH][OAc]. Dimethyl sulfoxide (DMSO) was used as a co-solvent to reduce the viscosity of the cellulose in the IL. Viscosities were within a comparable range: cellulose diacetate (CDA), cellulose laurate (CL), and cellulose palmitate (CP) have 13.2, 17.0, and 14.3 Pas, respectively [15]. The polymer solution containing 3.5 wt% cellulose was transesterified with derivatives of varying chain lengths—C2 (acetate), C12 (laurate), and C16 (Palmitate); for ease of reference throughout the study, they will be referred to as CDA, CL, and CP. The degrees of substitution (DS) for CL and CP were 1.1 and 0.8, respectively, while CDA had a DS of 2.4. The variation in DS was to ensure that the viscosities of the polymer solutions remained within a comparable range. The overall reaction yield was approximately 80%; however, due to multiple washing and filtration cycles, an exact yield value could not be determined. DS and yields are consistent with previously published articles [15,17].

#### 2.1.2. Solvent Casting of Cellulose Films

Cellulose ester solutions were prepared by dissolving cellulose esters in pyridine (5–7 wt.%) under continuous stirring at 60 °C for 16 h to ensure complete dissolution. The concentrations were based on achieving similar solution viscosities for the ester derivatives. Once dissolution was complete, the homogeneous solutions were cast onto a glass substrate. Film drying was then carried out using two different ovens—a forced air circulation oven (RO) and vacuum oven (VO)—to study the effects of the drying conditions.

The polymer solution was spread over a glass substrate using a 100 mm-wide film casting knife (BYK-Gardner GmbH, Germany) with a casting thickness of 100 mm [7,8,9,10]; the final film thickness was determined by the concentration of the polymer in the casting solution. The cast films were dried in the vacuum oven at 25 °C and at −0.45 bar overnight. After solvent evaporation, the films were immersed in distilled water to promote detachment from the substrate, then subjected to drying in the vacuum oven under the same conditions for 8 h. For films prepared in the air circulation oven, the same casting procedure was followed, with the samples dried at 25 °C overnight and immersed in distilled water to detach the films from the substrate, and dried in the oven for 8 h at 25 °C.

The drying parameters were selected based on the literature [15,18] to ensure complete solvent removal and reliable comparisons between different drying environments. The films produced were flexible and optically transparent (Figure 1).

### 2.2. Characterisation

The structural characterisation of the films was performed using X-ray diffraction (XRD) analysis on an Ultima IV diffractometer (Rigaku, Tokyo, Japan). The instrument was equipped with a silicon detector, Ni filter, and a Cu Kα irradiation source (*λ*  =  1.540 Å), anode voltage 40 kV, anode current 40 mA, *θ-θ* regime, step *θ*  =  0.02 deg. Copper anode Kα1 = 1.5406 Å without filtering Kα2Å. The Ni filter was applied for removing Kbeta. All analysis was made using Kα1. Kα2 was removed by an analysis programme.

Thermogravimetric analysis (TGA) was performed using a Setaram Labsys Evo 1600 thermoanalyzer (Setaram Instrumentation, Caluire, France). The DTG-DTA experiments were carried out under non-isothermal conditions up to 600 °C at the heating rate of 10 °C min^−1^ in the atmosphere of argon. Samples (7.2 ± 0.3 mg) were placed in a standard 100 μL alumina crucibles, with the argon gas flow rate being maintained at 20 mL min^−1^.

The surface microstructure of the cellulose ester films was examined using scanning electron microscopy (SEM) with a high-resolution Gemini Zeiss microscope (HR-SEM Zeiss Merlin, Oberkochen, Germany). The samples were cryo-fractured in liquid nitrogen and mounted vertically on a stub using a double-sided adhesive tape. A low vacuum detector was used in tandem with low vacuum mode (LVM), with water vapour kept at 90 pascal chamber pressure to examine the surface of the cellulose films using a Nova NanoSEM 450 (FEI—Thermo Fisher, Hillsboro, OR, USA) to reduce the possibility of distortion under the electron beam.

Atomic force microscopy (AFM) of the ester films was performed using a Bruker Dimension Edge (Bruker, Billerica, MA, USA) operating in Tapping mode. The surface topography measurements of the samples were taken for the area of 5 μm × 5 μm. The roughness average (Ra) and the root mean square (Rq) were calculated to quantitatively estimate the roughness of the films.

The contact angles (CA) were performed using the sessile drop method on a Data Physics OCA-20 (Riverside, CA, USA), under the controlled environmental conditions of a room temperature of 22 °C and a relative humidity of 65%. The deionised water drop is placed on the film surface from a micro syringe (Hamilton-Bonaduz). Measurements were taken at three different locations and averaged with the contact angles recorded over 40 s for each measurement.

## 3. Results

### 3.1. X-Ray Diffraction Analysis (XRD)

X-ray diffraction patterns of cellulose ester films reveal structural modification due to esterification. A broad peak at 2θ = 19.6 ° is observed in multiple samples, indicating an amorphous structure. The substitution of fatty acid side chains disrupts the ordering of cellulose, resulting in the amorphization of cellulose esters [19], or changes the structure to cellulose II polymorph [20]. The peak around 20° ascribed to the characteristic amorphous phase is also seen in the deconvolution curves in Figure 2. Similar deconvoluted diffractograms are observed for other studied cellulose esters. Based on the literature, drying rate and temperature influence the solvent’s evaporation kinetics and polymer chain mobility during solidification [21]. However, in our research, we did not see any influence of the drying method used on the sample structure; both methods showed nearly identical diffraction patterns, referenced in the Appendix A. Attachment of the long aliphatic side chains also prevents cellulose recrystallisation [11,12,13], which could explain the diffraction intensity of the peak around 20° [19,22,23]; this peak may also contain some crystal moieties, as suggested by some authors [24]. The peak around 2θ = 25–26° is unusual for native cellulose (cellulose I), which typically exhibits diffraction peaks at 15°, 16.5°, and 22.6 [25]. However, cellulose esters, which are usually derived from regenerated cellulose (cellulose II), exhibit different structural arrangements during esterification. The 25° peaks may reflect such modification or partial ordering induced by the ester side chain [26]. XRD patterns of cellulose esters, such as cellulose acetate, revealed shifted peaks near this region, suggesting altered structures [27].

### 3.2. Thermogravimetric Analysis (TGA)

All films were cast with identical casting areas and uniform thickness. Both vacuum-dried (VO) and oven-dried (RO) cellulose ester films had similar behaviour in relation to thermal degradation. In all the samples, the significant degradation peak of the cellulose backbone was observed at approximately 360 °C [28,29,30]. In the literature, similar degradation temperatures have been observed with functionalised celluloses, fatty acid chlorides [31]. The dynamic thermogravimetric curves of cellulose ester are given below in Figure 3. The initial decomposition temperature at 5% weight loss ($T_{5\%}$), the maximum weight loss temperature ($T_{d}$), temperature at 50% ($T_{50\%}$) weight loss and the char residue at 600 °C are listed in Table 1 below. ($T_{d1}$) is the initial step of pyrolysis of volatile compounds.

The main degradation range, between 250 and 370 °C, is due to the degradation of the cellulose bone [32]. Across all samples—CDA, CL, and CP—the T50% values for VO- and RO-dried films differed by less than 2%. These results show that the main degradation occurs at comparable temperatures regardless of drying techniques. The ester groups caused the formation of a specific mass loss in the range between 140 and 250 °C due to the decomposition of the alkyl chains that were grafted, which is seen in the TGA and dTG graph Figure 3. The cellulose esters CDA and CL indicate two step degradation; the similar trend is absent in the case of cellulose palmitate (CP); possibly, the alkyl chain that was grafted would have degraded along with the cellulose backbone [21]. It was observed that the drying processes studied did not make significant distinctions between the cellulose ester films and do not affect the thermal degradation. XRD analysis further supports this finding, revealing similar patterns among the ester films. The consistency between the XRD and TGA results suggest that the molecular arrangement and thermal stability of the cellulose ester films remained largely unaffected by the drying methods.

### 3.3. Scanning Electron Microscopy (SEM)

SEM was utilised to investigate the surface, internal morphology, and structural integrity of cellulose ester thin films prepared under varying drying conditions. The top view of the cellulose ester films revealed relatively flat, smooth surfaces and no surface porosity was observed on the films listed in the Appendix A. Comparative analysis of the cross-sectional images for VO and RO (Figure 4) revealed no notable structural differences, with no pores and layered structures. The observed structural homogeneity could be attributed to the controlled solvent evaporation, along with complete cellulose dissolution of the cellulose, resulting in a homogeneous polymer solution which prevents aggregate formation and minimises structural irregularities during film formation [33,34]. This observation is consistent with complete polymer dissolution, which, along with drying conditions, results in dense uniform films with a smooth morphology. However, the cross-sectional morphology of CDA films dried under VO and RO indicates a rough structure compared to CL and CP, respectively; the roughness may be attributed to solvent–polymer interaction during film formation.

### 3.4. Atomic Force Microscopy (AFM)

The surface morphology and the roughness values of the cellulose ester films are listed in Table 2. The darker areas, as observed, are associated with depressions on the surface of the films, and the lighter areas are the elevations. The surface morphology is characterised by root mean square (Sq), mean roughness (Sa), and the difference between high peaks and low valleys (Z). Observations from the AFM indicated the nodular structure. The polymer-rich phase has less ability to deform and thus merge and cause nodular structure due to chain entanglements [35]. These nodules are the influencing factor for the roughness of the surface. The effect of the evaporation environment is depicted clearly by comparing the surface images and the surface roughness parameters. Films dried under vacuum oven conditions demonstrated greater surface uniformity compared to those prepared under atmospheric pressure using a conventional oven [36]. VO drying enhances solvent removal, as solvents exhibit a lower boiling point under reduced pressure. This decrease in boiling point temperature can be leveraged to accelerate the diffusion of volatiles [37,38]. As the solvent evaporates, the cast solution becomes more concentrated, resulting in a decrease in the polymer diffusion coefficient. This means that as the solvent evaporates quickly (under vacuum), the rate at which the polymer materials diffuse to the film surface is slower. Consequently, there is more time for the materials to settle into a smooth uniform layer. During evaporation in the RO, solvent molecules at the film surface evaporate first, resulting in the formation of a polymer-rich layer at the top [39]. This polymer top layer acts as a barrier to further solvent evaporation, leading to uneven solvent removal. This disruption of polymer interactions by residual solvent molecules can lead to a non-uniform rough surface morphology of the resulting film [40,41].

The average Z-max depression values were observed to be higher in films dried under vacuum (VO) conditions compared to those dried in a regular oven (RO), as shown in Figure 5 for the CDA, CL, and CP samples. These variations are attributed to the differences in the air–solution interface’s behaviour during solvent evaporation [42]. The size and distribution of surface nodules are influenced by the drying environment, particularly the rate of solvent evaporation [42]. Prior studies indicate that an increase in the nodule size generally corresponds to greater roughness [43]. The roughness of the thin films can be influenced by the viscosity of the polymer solution. Despite maintaining viscosities within a comparable range—CDA, CL, and CP at 13.2, 17.0, and 14.3 Pas, respectively—the roughness varied across samples with varying drying conditions, highlighting that viscosity alone does not fully determine surface morphology. This is seen in the results presented in Table 2, where the CDA, CL, and CP films dried under the vacuum conditions (VO) show lower RMS and mean surface roughness values compared to those dried in a regular oven (RO). For CDA, VO decreased RMS from 8.12 (±1.9) nm to 6.55 (±0.3) and for CP, VO RMS decreased from 3.09 (±0.4) nm to 2.14 (±0.3) nm [44], and for CL, VO decreased roughness from 3.42 (±0.5) nm to 2.06 (±0.6) nm.

However, the nodular structure is seen on the surface of all films in both the vacuum and the regular oven. In this case, surface roughness cannot be in proportion to the nodule size, but the depression and the elevation regions could account for the surface roughness. The film surface indicates accumulated areas of elevation and depressions. The region of elevation is significant in oven-dried films compared to vacuum-dried films. The isolated nodules have distinct boundaries, and the interstitial regions can be clearly identified [45]. It could be possible that polymer chains that are present in the interstitial regions are randomly distributed compared to polymer chains that are present in the nodules [43]. However, a couple of samples indicate a circular concentration of peaks, which is attributed to the released air bubbles during the long drying process, which can be seen from Table 2. Overall, there is a drop in the surface roughness of the samples dried in VO compared to RO.

### 3.5. Contact Angle Measurement

The cellulose ester films were characterised by a static contact angle measurement to determine their hydrophobic or hydrophilic nature. It is well known that the long aliphatic chains are hydrophobic, and cellulose is hydrophilic. The results indicate an increased contact angle compared to our previous research, which had contact angle values of around 70° for CDA and 90° for CP [15], with about a 20% increase in the contact angles of the ester films, owing to the drying methods [15]. The contact angle of the films dried in the RO ranged between 80° and 121° (see Table 3), whereas for the VO-dried films, the contact angles ranged between 85° and 124°, depending on the fatty acid chain length. The films are hydrophobic if the contact angles are higher than 90° [46]. Nevertheless, the VO-dried cellulosic ester thin films exhibit higher contact angles as a result of the drying method, as evidenced by the results that the hydrophobicity of the smooth surface of the films dried in the vacuum oven is increased due to the low peaks and valleys on the film’s surface, which results in a reduced amount of contact between the droplet of water and the film surface. The contact angle measurement revealed that the contact angle between the water droplet and the film’s surface increases with an increase in alkyl chain length. The CDA ester with shorter alkyl chains had contact angle values ranging from 80 to 85°. In the case of the ester films with longer alkyl chains, the contact angle values ranged from 106 to 124° [47]. The hydrophobicity of long aliphatic side chains is well established, as evidenced by the high contact angles of cellulose palmitate and cellulose laurate.

### 3.6. Correlation of Cellulose Ester Films’ Surface Properties

The analysis of surface roughness and contact angle across the cellulose esters CDA, CL, and CP under two drying methods, RO and VO, reveals a consistent inverse relationship: as the surface roughness decreases, the contact angle increases, indicating enhanced hydrophobicity seen in Figure 6. This trend is especially pronounced under VO drying, where both the RMS and mean roughness values are generally lower, and the contact angles are higher in the RO. These results suggest that smoother surfaces, particularly those dried in the VO, tend to exhibit greater hydrophobicity in addition to the inherent characteristics of cellulose esters, based on their chain length. The regression analysis demonstrates a strong positive correlation (r = 0.952) between the contact angles and surface roughness values; however, it is not significant (*p* > 0.05), due to the high variation in the data. Based on the two tailed paired T-test, a notable difference between RO and VO for CDA (*p* = 0.199) with no statistical difference, while CL (*p* = 0.071) and CP (*p* = 0.018) show a significant statistical difference, as seen from the roughness values. Similarly, a contact angle comparison reveals no significant difference for CDA (*p* = 0.056), but CP (*p* = 0.023) and CL (*p* = 0.466) reflect a significant difference.

## 4. Conclusions

This study elucidates the influence of drying conditions on the structural, thermal, and surface properties of cellulose ester films. Structural analysis using XRD and thermal analysis using TGA revealed no noticeable difference between films dried using the VO and RO methods. Film produced via VO drying indicated enhanced surface characteristics, as evidenced by contact angle measurements and AFM. Specifically, esters with longer alkyl side chains demonstrated higher contact angles, indicative of increased hydrophobicity, alongside smoother surface morphologies. The 3D AFM images further confirmed the reduced surface roughness of VO-dried films. These results highlight the critical role of drying conditions in modulating surface properties without compromising bulk characteristics. Controlled drying, particularly under a vacuum, offers an approach to producing homogeneous, low roughness, and hydrophobic surfaces with applications in coatings, packaging, and biomedical materials, where film topology influences biological interactions. The ability to tune surface properties through drying conditions enables the design of cellulose-based films with a targeted performance. Looking ahead, these findings highlight VO drying as a promising, scalable route for tailoring surface functionalities in cellulose-based films, integrating this strategy into continuous processing systems.

## Figures and Tables

**Figure 1 polymers-17-03026-f001:**
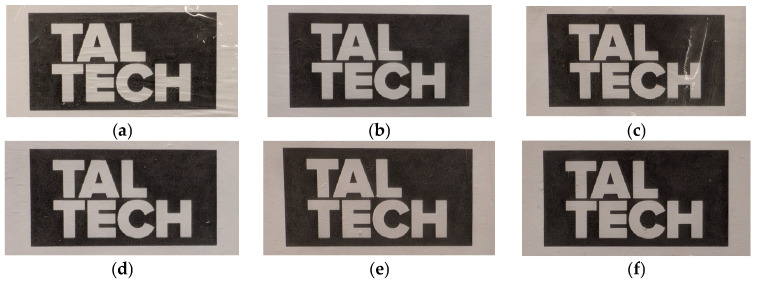
Colourless and transparent films of synthesised cellulose esters (**a**) CDA-VO, (**b**) CDA-RO, (**c**) CL-VO, (**d**) CL-RO, (**e**) CP-VO, and (**f**) CP-RO.

**Figure 2 polymers-17-03026-f002:**
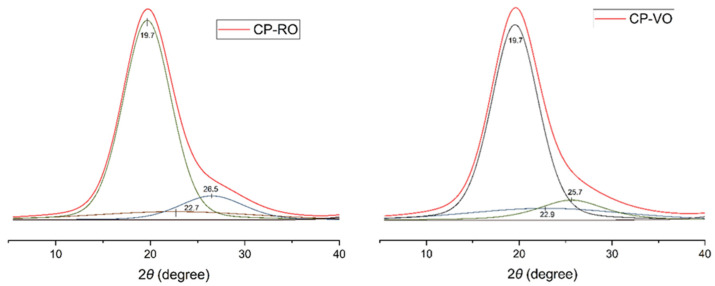
Deconvolution pattern of cellulose ester CP, dried in VO and RO.

**Figure 3 polymers-17-03026-f003:**
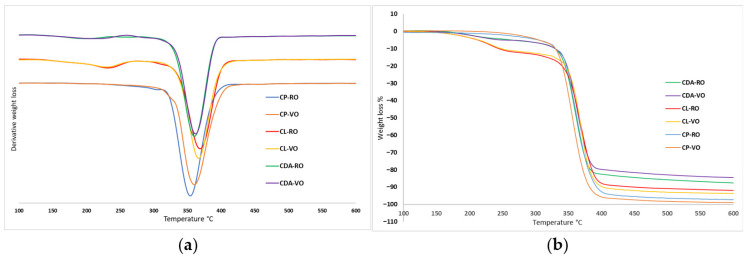
(**a**) DTG and (**b**) TGA curves of cellulose esters CL, CP, and CDA, comparing VO vs. RO.

**Figure 4 polymers-17-03026-f004:**
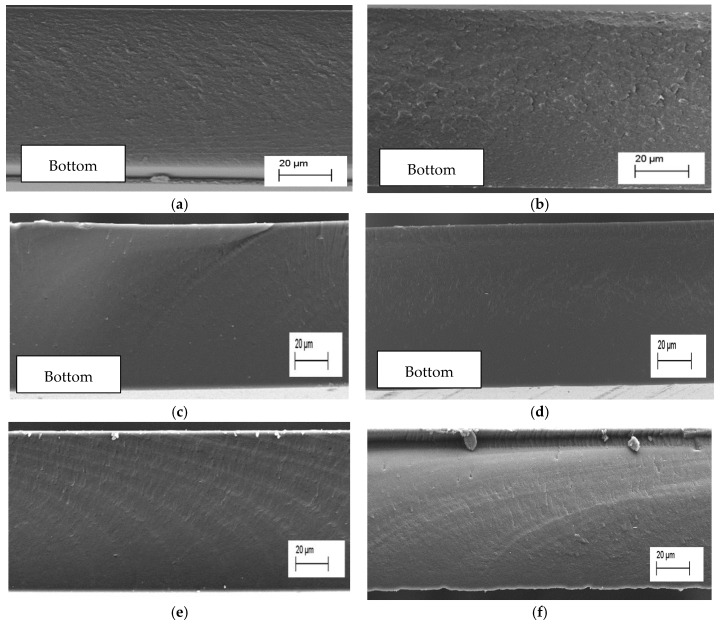
SEM cross-section images cellulose esters are magnification of 1000× (**a**) CDA-RO, (**b**) CDA-VO, (**c**) CL-RO, (**d**) CL-VO, (**e**) CP-RO, and (**f**) CP-VO.

**Figure 5 polymers-17-03026-f005:**
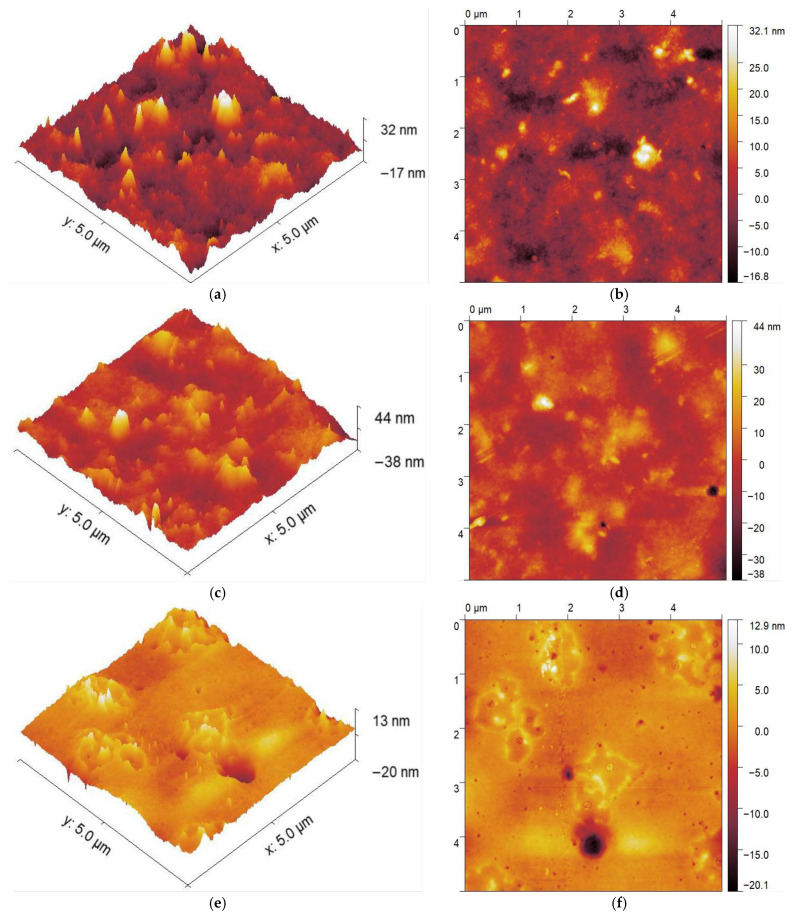
AFM images of cellulose esters: 3-D (**left**) and 2-D (**right**) images of (**a**,**b**) CDA-RO, (**c**,**d**) CDA-VO, (**e**,**f**) CL-RO, (**g**,**h**) CL-VO, (**i**,**j**) CP-RO, and (**k**,**l**) CP-VO.

**Figure 6 polymers-17-03026-f006:**
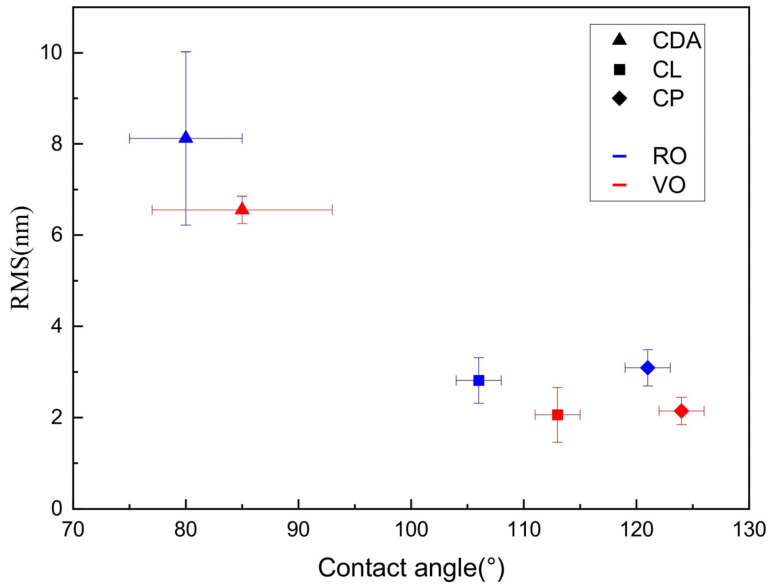
Correlation of contact angle and surface roughness of CDA, CL, and CP.

**Table 1 polymers-17-03026-t001:** Thermal stabilities of the cellulose fatty chain esters.

Sample		T_5%_ °C	T_d1_ °C	T_50%_ °C	T_d_ °C	Residue %
CDA	RO	263.2	201.1	362.4	360.8	12.2
	VO	245.0	210.6	363.6	362.0	15.3
CL	RO	212.5	232.0	366.7	369.2	7.9
	VO	212.7	230.8	365.7	367.3	6.3
CP	RO	303.4	-	362.2	360.6	2.6
	VO	305.0	-	354.4	354.4	0.9

**Table 2 polymers-17-03026-t002:** AFM surface roughness values of the cellulose fatty chain esters.

Sample	RMS Roughness (Sq) (nm)		Mean Roughness (Sa) (nm)	
Drying Method	RO	VO	RO	VO
CDA	8.12 (±1.9)	6.55 (±0.3)	5.83 (±1.3)	4.71 (±0.1)
CL	3.42 (±0.5)	2.06 (±0.6)	2.32 (±0.3)	1.27 (±0.3)
CP	3.09 (±0.4)	2.14 (±0.3)	2.30 (±0.3)	1.66 (±0.2)

**Table 3 polymers-17-03026-t003:** Average contact angle measurement of cellulose ester films.

Ester	RO (°)	VO (°)
CDA	80 (±5)	85 (±8)
CL	106 (±2)	113 (±3)
CP	121 (±2)	124 (±3)

## Data Availability

The data presented in this study are available upon request from the corresponding author. The data is not publicly available due to the confidentiality of the running project.

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
