# Peer review of "Effect of Drying Methods on the Morphological and Functional Properties of Cellulose Ester Films"

_polymers, 2025, doi:10.3390/polym17223026_

Round 1

Reviewer 1 Report (Previous Reviewer 2)

Comments and Suggestions for Authors

The experimental article "Effect of drying methods on the morphological and functional properties of cellulose ether films", devoted to the description of the effect of two drying methods of solutions long-chain fatty cellulose ethers on the properties of films, is a new version of the revised but not approved manuscript of the authors. By all formal criteria, the manuscript corresponds to the publication in the journal "Polymers", since it is devoted to polymers — cellulose ethers — and reflects all stages of the study, from the production of esters to the description of the properties of films obtained from them. The scientific hypothesis of the authors is to discuss the effect or absence of the effect of drying methods on the properties of films. The authors' results are consistent with classical ideas and fundamental knowledge in the field of cellulose ethers. This manuscript is a continuation of many years of systematic studies of cellulose ether films conducted by this group of authors. What is especially important, the authors explicitly state that there is no effect of drying methods on the crystal structure of amorphous films of cellulose ethers and on their thermal stability. Drying methods determine only changes in the surface of the film, which is quite natural, but the manuscript describes a specific case with two experimental cellulose ethers and one industrial ether for comparison. I believe that the manuscript in its present form may be of interest to readers who study methods for obtaining films with desired properties.

There are no comments on the design of the manuscript: figures and tables are necessary in this case, although there are also Supplementary.

I recommend publishing it as is. 

Author Response

Thank you!

Reviewer 2 Report (New Reviewer)

Comments and Suggestions for Authors

The article investigates how different drying methods, such as vacuum oven and regular air oven, influence the morphology, structure, and surface properties of cellulose ester films. Techniques used include XRD, TGA, SEM, AFM, and contact angle measurements. The main finding is that while internal structure and thermal stability are largely unaffected, surface smoothness and hydrophobicity improve under vacuum drying. The use of a novel ionic liquid and the comparison of drying methods add originality. This article can be accepted before improving some issues.

1) This work presents some grammar mistakes, for example, occasional tense inconsistencies, switching between present and past. Please fix them.

2) In results and discussion, the discussion could be expanded by explaining why vacuum drying leads to smoother surfaces, potentially relating to evaporation kinetics or solvent diffusion rates.

3) Figure 2 appears twice (once for XRD, once for TGA); the numbering should be sequential.

4) Reference 11 and 12 are identical (both Wang et al., 2022).

Comments on the Quality of English Language

Please see the report.

Author Response

1) This work presents some grammar mistakes, for example, occasional tense inconsistencies, switching between present and past. Please fix them.

              Thank you very much for pointing that out. I gave carefully reviewed the work to correct the tense inconsistencies and other grammatical mistakes.

2) The discussion could be expanded by explaining why vacuum drying leads to smoother surfaces, potentially relating to evaporation kinetics or solvent diffusion rates.

Thank you for bringing this to our attention. Films dried under vacuum oven conditions demonstrated greater surface uniformity compared to those prepared under atmospheric pressure using a conventional oven[40], VO drying enhances solvent removal, as solvents exhibit a lower boiling point under reduced pressure. This decrease in boiling point temperature can be leveraged to accelerate the diffusion of volatiles[41,42] As the solvent evaporates, the cast solution becomes more concentrated, resulting in a decrease of polymer diffusion coefficient. This means that as the solvent evaporates quickly (under vacuum), the rate at which the polymer materials diffuse to the film surface is slower. Consequently, there is more time for the materials to settle into a smooth uniform layer. During evaporation in the RO, solvent molecules at the film surface evaporate first, resulting in the formation of a polymer rich layer at the top[43] This polymer top layer acts as a barrier to further solvent evaporation, leading to uneven solvent removal. This disruption of polymer interactions by residual solvent molecules can lead to non-uniform rough surface morphology of the resulting film[44,45].

3) Figure 2 appears twice (once for XRD, once for TGA); the numbering should be sequential.

              Thank you for bringing this to our attention. The Figure 2 has been updated so that all the figures now follow a proper sequential order.

4) Reference 11 and 12 are identical (both Wang et al., 2022).

              Thank you for noticing that mistake. I have corrected the references and any duplicates have been resolved.

This manuscript is a resubmission of an earlier submission. The following is a list of the peer review reports and author responses from that submission.

Round 1

Reviewer 1 Report

Comments and Suggestions for Authors

The study investigates: “Effect of Drying Methods on the Morphological and Functional Properties of Cellulose Ester Films”

The authors need to address the following comments:

  • The abstract does not discuss the results thoroughly and the authors are required to re-write the abstract and the main findings have to be included in the discussion.
  • Line 29 to 40 in the introduction section, cannot have a single reference, there is a need to add more references. For example, if the authors suggest cellulose is not thermoplastic, it has to be supported by a reference(s).
  • There is also limited literature in the introduction in relation to the current study
  • The novelty as well, has to come very strong from the introduction section
  • In section 2.1.2, expand more on the methods than providing references.
  • What is the main reason for drying cellulose @105 degrees, what could be the reason.
  • One of the suggestions is that the authors are advised to run FTIR to verify the esterification with cellulose.
  • Furthermore, the authors are advised to plot the normal TGA graphs and discuss them together with the DTG
  • It is advised that T50% be calculated as well.
  • The conclusion may be modified further, with the addition of future trend.

Author Response

I thank the editor for taking time to review the article and all the comments made and pointing out the corrections needed. The corresponding corrections/revisions are presented in the resubmitted files.

Q1- The abstract does not discuss the results thoroughly and the authors are required to re-write the abstract and the main findings have to be included in the discussion.

This study presents the synthesis and characterisation of cellulose long chain fatty acid ester films using a novel distillable ionic liquid (IL),  5-methyl-1,5,7-triaza-bicyclo-[4.3.0] non-6-enium acetate [mTBNH][OAc] in combination with DMSO as a cosolvent. Cellulose esters- cellulose diacetate (CDA), cellulose laurate (CL), and cellulose palmitate (CP) were fabricated through evaporation induced phase separation method (EIPS) and dried under two conditions: conventional oven drying (RO) and vacuum oven drying (VO). The influence of drying conditions on the structural, thermal, and surface properties of the films was evaluated using XRD,TGA,SEM, AFM, and contact angle measurement techniques.  XRD confirmed an amorphous structure in all films, with no significant effect on drying conditions. TGA revealed consistent thermal degradation profiles across all samples, with ester group decomposition accruing between 140 - 250 °C and main cellulose backbone degradation near 350 °C. SEM cross section showed uniform film devoid of cavities and layered structures. AFM analysis demonstrated that VO dried films had smoother surfaces compared to RO dried films, correlating with increased contact angles and enhanced hydrophobicity. A strong inverse relationship between surface roughness and hydrophobicity was observed particularly in VO dried samples, although not statistically  significant due data variability. Overall, the drying method had minimal impact on the internal structure and thermal stability, it significantly influenced surface morphology and wettability.

Q2- Line 29 to 40 in the introduction section, cannot have a single reference, there is a need to add more references. For example, if the authors suggest cellulose is not thermoplastic, it has to be supported by a reference(s).

               Additional references are added

Q3- There is also limited literature in the introduction in relation to the current study.

               The solution casting technique is a simple yet reliable approach for film fabrication, where the polymer solution is uniformly dispensed onto a substrate, followed by solvent evaporation under controlled or partially controlled conditions to yield films. Rapid solvent evaporation during film formation has garnered increasing attention due to its operational simplicity and cost effectiveness In such methods, solutions of hydrophobic polymers are subjected to accelerated solvent evaporation, resulting in the formation surface morphologies. The resulting topological features are closely associated with enhanced water repellency. The impact of experimental drying parameters on film properties (temperature, pressure and etc.) such as wettability and topography, has been explored minimally in the literature.

Studies have reported that drying conditions have shown to significantly influence the morphology, optical appearance, and functional properties of cellulose based films. Lyytikäinen et al. (2021) demonstrated that drying temperature affects wettability and barrier performance, while substrate roughness can induce air entrapment, altering surface morphology.  Harini and Sukumar (2019) observed that vacuum drying yields transparent cellulose acetate films, whereas ambient drying results in opacity, highlighting the critical role of drying environment. These findings emphasize the importance of controlling drying parameters to optimize film properties.

Q4- The novelty as well, has to come very strong from the introduction section

               this study tests the hypothesis that drying conditions significantly influence the physical properties of long chain cellulose ester films. By systematically comparing vacuum oven (VO) and forced air circulation (RO) drying at elevated temperatures, the research investigates how solvent evaporation dynamics affect film morphology, thermal stability, and surface characteristics. It is poised that the drying environments governs film formation mechanisms, thereby impacting the uniformity and performance of the resulting materials. Comparison with previously reported room temperature air dried films provide further insight into how the drying parameters effect the morphology, structure, thermal, and wetting properties.

Q5- In section 2.1.2, expand more on the methods than providing references.

The commercial cellulose obtained from Carl Roth Gmbh & Co. Kg, Karlsruhe, Germany (CAS no. 9004-34-6) and has a density of 1.5 g/cm3, a pH value of 5–7, and a fibre length of 0.02–0.1 mm. Molar mass (MM) of pure cellulose was determined at 25 °C from the intrinsic viscosity [η] of cellulose solution in cupriethylenediamine hydroxide, CuEn, according to a standard procedure ASTM D1795-13. The MM was then calculated by the Mark-Houwink equation with parameters K = 1.01 × 10−4 dL/g and a = 0.9. The obtained MM was 163,000 g/mol. The cellulose was used for the transesterification process with vinyl esters in ionic liquid (IL), i.e., 5-methyl-1,5,7-triaza-bicyclo-[4.3.0]non-6-enium acetate, [mTBNH][OAc], which was synthesized by Liuotin Group Oy, Taramäentie, Finland. The viscosity of cellulose in IL was reduced using dimethyl sulfoxide (DMSO) as a co-solvent. Vinyl esters with a purity >98% were purchased from Tokyo Chemical Industry Co. (Tokyo, Japan). DMSO was purchased from Fisher Chemical, Waltham, MA, USA (CAS no. 67-68-5) and pyridine with a purity of ≥99.5% was acquired by Sigma-Aldrich, St. Louis, MO, USA (CAS no. 7291-22-7). Viscosities were within a comparable range, CDA, CL, and CP at 13.2, 17.0, and 14.3 Pas respectively.

Q6- What is the main reason for drying cellulose @105 degrees, what could be the reason.

               The reason for drying cellulose is to remove any remaining moisture.

Q7- One of the suggestions is that the authors are advised to run FTIR to verify the esterification with cellulose.

               FTIR data was omitted because the results were similar to those of the articles previously published with similar method of cellulose transesterification.

Q8- Furthermore, the authors are advised to plot the normal TGA graphs and discuss them together with the DTG.  It is advised that T50% be calculated as well. It is advised that T50% be calculated as well.

               The graph of TGA is added and discussed.

Q9- The conclusion may be modified further, with the addition of future trend.

               Based on the findings, it would be recommended that future work involving solvent casting techniques explore scalability of drying conditions and its impact on application specific performance, advancing the development of tailored functional materials.

Reviewer 2 Report

Comments and Suggestions for Authors

The experimental article "Effect of Drying Methods on the Morphological and Functional Properties of Cellulose Ester Films" covers a very relevant topic of obtaining high-quality films from cellulose ethers, while, as is evident from the title, "morphology and functional properties depend on drying methods." Unfortunately, neither the abstract nor the conclusions of the article explain how the morphology and properties depend on drying methods. The opposite effect is described: "no effect." For example, the abstract states that "Different drying conditions have little to no effect on the crystal structure" and "The thermal stability of the films...were not affected by the varying drying conditions." For this reason alone, the authors are advised to think about the formulation of the scientific result of the studies, after which the simplest thing to do is to change the title. Given the fundamental nature of the Polymers publication, it will be difficult for the authors to convince readers that the drying methods of finished cellulose ethers will affect the natural properties of the target products. It is also necessary to overcome the decrease in the value of "Percent match: 39%" to 15%. Other recommendations are given below in the list.

Recommendations:

  1. Introduction. It is recommended to list the factors and give examples that determine the morphological and functional properties of cellulose ether films.
  2. Lines 82-89. It is recommended to present a scientific hypothesis for a specific cellulose ether, since in the presented form the thesis suggests extremely wide possibilities.
  3. In section 2.1. Materials, it is recommended to provide some data on "fibrous cellulose", for example, its purity indicators and the degree of polymerization (or viscosity).
  4. In section 2.1.2. Cellulose transesterification, it is recommended to provide data on the obtained cellulose ethers: yield, degree of substitution, viscosity, etc.
  5. Lines 161-168. Since very little space was devoted to the fact of decreasing high crystallinity of cellulose after esterification in the introduction, it is recommended to describe the discussion of low-crystalline cellulose ethers more correctly. It is clear that it was not the drying method that reduced the degree of crystallinity of the resulting product. The section should contain the values ​​of the degree of crystallinity of cellulose ethers in comparison with the degree of crystallinity of the original.
  6. For what reason are only two samples of cellulose ethers discussed in section 3.1. X-ray diffraction analysis (XRD). Where are the others? Photo 1 shows six films.
  7. In section 3.2. Thermogravimetric Analysis (TGA), the results are discussed without specifying the provision of uniform film thickness. This phenomenon should be commented on by the authors.
  8. Lines 207-208. It is recommended to check the correctness of the sentence "The observed structural homogeneity can be attributed to the controlled solvent evaporation and supports the complete dissolution of cellulose", since cellulose ethers were dissolved.
  9. Comparison of the photographs in Fig. 2 demonstrates the difference between the CDA control film and the experimental films. The authors should discuss this difference.
  10. It is recommended to clarify directly for Section 3.4. Atomic Force Microscopy (AFM) that all the films under study were prepared from solutions of the same concentration of cellulose ester, ensuring the same film thickness. Or is this not the case? The results in Table 2 are different and require explanation.
  11. Rows 282-285. There is information that the films under study had different thicknesses (specifically, "thin").
  12. It is recommended to be more clear in the discussion of Table 3: does the drying method affect the average contact angle?
  13. In Section 3.7. Correlation of cellulose esters films surface properties, it is recommended to continue the discussion of the observed difference between the films and clearly state that this difference is due to the nature of the cellulose ester or the drying method used to prepare the film.
  14. Taking into account the above recommendations, Section 4. Conclusions needs to be rethought and rewritten with the statement of the proof of achievement of the stated goal at the beginning of the article. In the process, it is recommended to list the properties of the films in the same order in which they were studied. "Complementary analyses using XRD and TGA" are given in the article at the beginning, as fundamental, not additional. On the same topic: XRD and TGA data do not depend on the drying method, and the authors claim the opposite in line 324: "the critical role of drying conditions in modulating surface properties".
  15. What is the practical significance of the obtained results?
  16. References 7 and 9, as well as 27 and 28 are the same. It is necessary to check the entire list of references and strengthen the relevance of the first five references with more recent publications.
  17. Line 169 says: "Figure S1-S2". Check it is correct.

Author Response

I thank the editor for taking time to review the article and all the comments made and pointing out the corrections needed. The corresponding corrections/revisions are presented in the resubmitted files.

Q1- Introduction. It is recommended to list the factors and give examples that determine the morphological and functional properties of cellulose ether films.
               the required factors are mentioned

Q2- Lines 82-89. It is recommended to present a scientific hypothesis for a specific cellulose ether, since in the presented form the thesis suggests extremely wide possibilities.

               this study tests the hypothesis that drying conditions significantly influence the physical properties of long chain cellulose ester films. By systematically comparing vacuum oven (VO) and forced air circulation (RO) drying at elevated temperatures, the research investigates how solvent evaporation dynamics affect film morphology, thermal stability, and surface characteristics. It is poised that the drying environments governs film formation mechanisms, thereby impacting the uniformity and performance of the resulting materials. Comparison with previously reported room temperature air dried films provide further insight into how the drying parameters effect the morphology, structure, thermal, and wetting properties.

Q3- In section 2.1. Materials, it is recommended to provide some data on "fibrous cellulose", for example, its purity indicators and the degree of polymerization (or viscosity).

               the fibre length from the technical data sheet is added to the section of 2.1

Q4- In section 2.1.2. Cellulose transesterification, it is recommended to provide data on the obtained cellulose ethers: yield, degree of substitution, viscosity, etc.

               Viscosities were within a comparable range, CDA, CL, and CP at 13.2, 17.0, and 14.3 Pas respectively

Q5- Lines 161-168. Since very little space was devoted to the fact of decreasing high crystallinity of cellulose after esterification in the introduction, it is recommended to describe the discussion of low-crystalline cellulose ethers more correctly. It is clear that it was not the drying method that reduced the degree of crystallinity of the resulting product. The section should contain the values ​​of the degree of crystallinity of cellulose ethers in comparison with the degree of crystallinity of the original.

               Thank you for the for the comment. Since the cellulose used in this study was fully amorphous. The peaks that were shown was the overlap of the reordered structure and that is why the crystallinity peaks weren’t indicated.

Q6- For what reason are only two samples of cellulose ethers discussed in section 3.1. X-ray diffraction analysis (XRD). Where are the others? Photo 1 shows six films.

               The remaining images are added in the supplementary data.

Q7- In section 3.2. Thermogravimetric Analysis (TGA), the results are discussed without specifying the provision of uniform film thickness. This phenomenon should be commented on by the authors.

               All films were cast with identical casting areas and uniform thickness. Both vacuum dried (VO) and oven dried (RO) cellulose ester films have similar behaviour in relation to thermal degradation.

Q8- Lines 207-208. It is recommended to check the correctness of the sentence "The observed structural homogeneity can be attributed to the controlled solvent evaporation and supports the complete dissolution of cellulose", since cellulose ethers were dissolved.

               The observed structural homogeneity can be attributed to the controlled solvent evaporation along with complete cellulose dissolution of cellulose, resulting in a homogeneous polymer solution which prevents aggregate g formation and minimizing structural irregularities during film formation.

Q9- Comparison of the photographs in Fig. 2 demonstrates the difference between the CDA control film and the experimental films. The authors should discuss this difference

               However, the cross-sectional morphology of CDA films dried under VO and RO indicate a rough structure compared to CL and CP respectively, the roughness may be attributed to solvent polymer interaction during film formation.

Q10- It is recommended to clarify directly for Section 3.4. Atomic Force Microscopy (AFM) that all the films under study were prepared from solutions of the same concentration of cellulose ester, ensuring the same film thickness. Or is this not the case? The results in Table 2 are different and require explanation.

               This is seen in the results presented in the Table 2, where CDA and CP films dried under vacuum conditions (VO) show lower RMS and mean surface roughness values compared to those dried in a regular oven (RO). For CDA, VO decreased RMS from 8.12(±1.9)nm to 6.55(±0.3) and for CP, VO RMS decreased from 3.09(±0.4)nm to 2.14(±0.3)nm

               Although, a couple of samples indicate circular concentration of peaks which is attributed to the released air bubbles during the long drying process can be seen from Table 2, there is a drop in the surface roughness of the samples dried in VO compared to RO, except for CL where VO increased roughness from 3.42(±0.5) nm to 5.50(±1.1)nm which can be attributed to the micro defects during drying process possibly due to air bubbles.

Q11- Rows 282-285. There is information that the films under study had different thicknesses (specifically, "thin").

               The films had comparatively similar thicknesses.

Q12- It is recommended to be more clear in the discussion of Table 3: does the drying method affect the average contact angle?

               the VO-dried cellulosic ester thin films exhibit higher contact angles as a result of the drying method, as evidenced by the results that the hydrophobicity of the smooth surface of the films dried in the vacuum oven is increased due to the low peaks and valleys on the film surface, which results in a reduced amount of contact between the droplet of water and the film surface             

Q13- In Section 3.7. Correlation of cellulose esters films surface properties, it is recommended to continue the discussion of the observed difference between the films and clearly state that this difference is due to the nature of the cellulose ester or the drying method used to prepare the film.

               These results suggest that smoother surfaces, particularly those dried in the VO, tend to exhibit greater hydrophobicity in addition to the inherent characteristics of cellulose esters based on their chain length.

Q14- Taking into account the above recommendations, Section 4. Conclusions needs to be rethought and rewritten with the statement of the proof of achievement of the stated goal at the beginning of the article. In the process, it is recommended to list the properties of the films in the same order in which they were studied. "Complementary analyses using XRD and TGA" are given in the article at the beginning, as fundamental, not additional. On the same topic: XRD and TGA data do not depend on the drying method, and the authors claim the opposite in line 324: "the critical role of drying conditions in modulating surface properties".

               This study elucidates the influence of drying conditions on the structural, thermal and surface properties of cellulose ester films. Structural analysis using XRD and thermal analysis using TGA revealed no noticeable difference between films dried using VO and RO methods. film produced via VO drying indicated enhanced surface characteristics, as evidenced by contact angle measurements and AFM. Specifically, esters with longer alkyl side chain esters demonstrated higher contact angles, indicative of increased hydrophobicity, alongside smoother surface morphologies. The 3D AFM images further confirmed the reduced surface roughness of VO dried films. These results highlight the critical role of drying conditions in modulating surface properties without compromising bulk characteristics. Controlled drying, particularly under vacuum, offers an approach to producing homogeneous, low roughness, and hydrophobic surfaces with applications in coatings, packaging and biomedical materials where film topology influence biological interactions. The ability to tune surface properties through drying conditions enables the design of cellulose-based films with targeted performance. Looking ahead, these finding highlight VO drying as a promising, scalable route for tailoring surface functionalities in cellulose-based films, integrating this strategy into continuous processing systems.

Q15- What is the practical significance of the obtained results?

               The practical significance of the obtained results lies in the demonstrated ability to precisely tailor surface properties of cellulose ester films through simple modification of the drying process, without affecting their structural or thermal integrity.

Q16- References 7 and 9, as well as 27 and 28 are the same. It is necessary to check the entire list of references and strengthen the relevance of the first five references with more recent publications.

               The mentioned changes with references were made.

Q17- Line 169 says: "Figure S1-S2". Check it is correct.

               Figure S1-S2 were referring to the figures in the supplementary data.

Reviewer 3 Report

Comments and Suggestions for Authors

1. The paper studies the effect of drying method on film morphology, which is a good research topic, but the author sets few experimental parameters, especially important parameters such as temperature and period are not discussed, which makes the importance of the paper not fully reflected.
2. In terms of experimental details 2.1.3, it is recommended to add specific parameters of the oven, the solution volume and other details to make the experiment repeatable.
3. The author only focuses on the morphology of the film, and the morphological changes of the film in the experimental results are not so obvious. How about the mechanical, barrier properties, etc., especially those related to packaging films?

Author Response

I thank the editor for taking time to review the article and all the comments made and pointing out the corrections needed. The corresponding corrections/revisions are presented in the resubmitted files.

Q1. The paper studies the effect of drying method on film morphology, which is a good research topic, but the author sets few experimental parameters, especially important parameters such as temperature and period are not discussed, which makes the importance of the paper not fully reflected.

The final film thickness was dictated by the concentration of the polymer in the solution. Films were cast on laminated glass plates using the film casting knife width of 100 mm (BYK-Gardner GmbH, Germany). The parameters for drying were selected from the limited literature to ensure complete evaporation and enable a reliable comparison of the drying environments.

Q2-  In terms of experimental details 2.1.3, it is recommended to add specific parameters of the oven, the solution volume and other details to make the experiment repeatable.

               A knife blade was used to spread the polymer solution over a flat glass substrate of 100 mm width. The cast solutions were placed at 25 °C and at -0.45 bar in a vacuum oven over night. After removal from the oven the cast thin films were immersed in distilled water, peeled from the substrate, and further dried in the VO at 25°C with -0.45 bar for 8 hours. Similarly, the samples dried in the RO had been cast on a glass substrate overnight at 25°C, were immersed in distilled water and peeled from substate followed by drying in the RO for 8 hours at 25°C.

Q3- The author only focuses on the morphology of the film, and the morphological changes of the film in the experimental results are not so obvious. How about the mechanical, barrier properties, etc., especially those related to packaging films?

               While changes were moderate, they impact surface functionalities. Mechanical and barrier properties, crucial for packaging, will possibly be addressed in future work.

Round 2

Reviewer 1 Report

Comments and Suggestions for Authors

Accept in current form, all the comments were addressed.

Author Response

Thank you for your valuable comments and feedback

Reviewer 2 Report

Comments and Suggestions for Authors

Dear authors, You have made significant changes to the abstract of your experimental article, as well as minor changes to the text, but please note that the Percent match value has decreased slightly, from 39 to 36%. You will have to make changes to the main text of the article to reduce the value to 15%.

In addition, some of my questions and recommendations remained unanswered. Please answer or explain your inability to answer the recommendations. I leave the recommendation number unchanged: as stated in the first round.

  1. In section 2.1.2. Cellulose transesterification, it is recommended to provide data on the obtained cellulose ethers: yield, degree of substitution, viscosity, etc.

The updated version has three viscosity values. And nothing more. Your experimental data were compared with commercial cellulose acetate with a known degree of substitution. Since your experiment used a classic brand of cotton cellulose with a degree of polymerization of 1000, a question arises about the degree of esterification and (or) the yield of the ethers you obtained. Are these data missing? Most likely not. Provide them in the article.

  1. Lines 161-168. Since very little space was devoted to the fact of decreasing high crystallinity of cellulose after esterification in the introduction, it is recommended to describe the discussion of low-crystalline cellulose ethers more correctly. It is clear that it was not the drying method that reduced the degree of crystallinity of the resulting product. The section should contain the values of the degree of crystallinity of cellulose ethers in comparison with the degree of crystallinity of the original.

This question was asked because the reviewer, as well as future readers, do not understand at all the purpose of discussing the effect of drying on the crystallinity of amorphous cellulose ethers. This discussion remains in the updated version. It is recommended to take care of this topic now: clearly express the idea of the amorphous nature of ethers in contrast to crystalline cellulose and not promise readers a description of the effect of drying on crystallinity. Please make corrections to the text of the article again.

  1. For what reason are only two samples of cellulose ethers discussed in section 3.1. X-ray diffraction analysis (XRD). Where are the others? Photo 1 shows six films.

Indicate in lines 189-190 that everything else is in Salimentary.

  1. In Section 3.7. Correlation of cellulose esters films surface properties, it is recommended to continue the discussion of the observed difference between the films and clearly state that this difference is due to the nature of the cellulose ester or the drying method used to prepare the film.

In this case, it is necessary to provide data on the "chain length" in the article. Please follow this recommendation.

Author Response

Q1- In section 2.1.2. Cellulose transesterification, it is recommended to provide data on the obtained cellulose ethers: yield, degree of substitution, viscosity, etc.

               The updated version has three viscosity values. And nothing more. Your experimental data were compared with commercial cellulose acetate with a known degree of substitution. Since your experiment used a classic brand of cotton cellulose with a degree of polymerization of 1000, a question arises about the degree of esterification and (or) the yield of the ethers you obtained. Are these data missing? Most likely not. Provide them in the article.

Fibrous cellulose (CAS No. 9004-34-6) was procured from Carl Roth GmbH & Co. KG (Germany) with the average fibre length between 0.02 and 0.1mm, density of 1.5 g/cm3, and a pH range of 5-7. Following the ASTM D1795-13 standard, the average molar mass(MM) of cellulose was determined using intrinsic viscosity [η] measurement in cupriethylenediamine hydroxide (CuEn), at 25°C. Subsequently using Mark-Houwink equation with parameters K = 1.01 × 10−4 dL/g and a = 0.9 [17], yielding a molar mass value 163,000 g/mol. Trasnsesterification reaction of this cellulose was carried out with vinyl esters, using ionic liquid (IL), i.e., 5-methyl-1,5,7-triaza-bicyclo-[4.3.0]non-6-enium acetate, [mTBNH][OAc], which was synthesized by Liuotin Group Oy, Taramäentie, Finland. Dimethyl sulfoxide (DMSO) was as a co-solvent to reduce the viscosity of the cellulose in the IL. Viscosities were within a comparable range, cellulose diacetate (CDA-C2), cellulose laurate (CL-C12), and cellulose palmitate (CP-C16) at 13.2, 17.0, and 14.3 Pas respectively[16]. These derivatives with fatty acid chain lengths of C2 (acetate), C12 (laurate), and C16 (Palmitate), for ease of reference throughout the study, they will be referred to as CDA, CL, and CP. The polymer solutions were prepared at 3.5 wt% cellulose, with the degree of polymerisation (DP) of CL and CP ranging between 0.8 – 1.1 and CDA with DP of 2.4. The variation in DP was to ensure that the viscosities of the polymer solutions remained with in a comparable range. The DS and yields are consistent with previously published articles[16,17].

Q2- Lines 161-168. Since very little space was devoted to the fact of decreasing high crystallinity of cellulose after esterification in the introduction, it is recommended to describe the discussion of low-crystalline cellulose ethers more correctly. It is clear that it was not the drying method that reduced the degree of crystallinity of the resulting product. The section should contain the values of the degree of crystallinity of cellulose ethers in comparison with the degree of crystallinity of the original.

               This question was asked because the reviewer, as well as future readers, do not understand at all the purpose of discussing the effect of drying on the crystallinity of amorphous cellulose ethers. This discussion remains in the updated version. It is recommended to take care of this topic now: clearly express the idea of the amorphous nature of ethers in contrast to crystalline cellulose and not promise readers a description of the effect of drying on crystallinity. Please make corrections to the text of the article again.

X-ray diffraction patterns of cellulose ester films reveal structural modification due to esterification. A broad peak at 2θ = 19.6 ° is observed multiple samples, indicating an amorphous structure. The substitution of fatty acid side chains disrupts the ordering of cellulose, resulting the amorphization of cellulose esters [20] or change the structure to cellulose II polymorph.

Q3-  For what reason are only two samples of cellulose ethers discussed in section 3.1. X-ray diffraction analysis (XRD). Where are the others? Photo 1 shows six films.

Indicate in lines 189-190 that everything else is in Salimentary.

               However, in our research, we did not see any influence of drying method used on the sample structure, both methods showing nearly identical diffraction pattern referenced in the supplementary data as Figure S1-S2.

Q4- In Section 3.7. Correlation of cellulose esters films surface properties, it is recommended to continue the discussion of the observed difference between the films and clearly state that this difference is due to the nature of the cellulose ester or the drying method used to prepare the film.

In this case, it is necessary to provide data on the "chain length" in the article. Please follow this recommendation.

               The viscosities were addressed in section 2.1.2

Viscosities were within a comparable range, cellulose diacetate (CDA-C2), cellulose laurate (CL-C12), and cellulose palmitate (CP-C16) at 13.2, 17.0, and 14.3 Pas respectively[16]. These derivatives with fatty acid chain lengths of C2 (acetate), C12 (laurate), and C16 (Palmitate), for ease of reference throughout the study, they will be referred to as CDA, CL, and CP. The polymer solutions were prepared at 3.5 wt% cellulose, with the degree of polymerisation (DP) of CL and CP ranging between 0.8 – 1.1 and CDA with DP of 2.4. The variation in DP was to ensure that the viscosities of the polymer solutions remained with in a comparable range. The DS and yields are consistent with previously published articles.

Reviewer 3 Report

Comments and Suggestions for Authors

Concerning the surface functionalities, it's better to add the significance analysis of the corresponding data to compare the effect of RO and VO.

Author Response

Q1- Concerning the surface functionalities, it's better to add the significance analysis of the corresponding data to compare the effect of RO and VO.

The regression analysis demonstrates a strong positive corelation (r = 0.952) between the contact angles and surface roughness values, however it was not significant (p > 0.05) due to high variation in data. Based on the two tailed paired T-test, a notable difference be-tween RO and VO for CDA (p = 0.199) with no statistical difference, while CL (p = 0.008) and CP (p = 0.018) show significant statistical difference, seen from the roughness values. Similarly, contact angle comparison reveal no significant difference for CDA (p = 0.056) but CP (p = 0.023) and CL (p = 0.466) reflects a significant difference.

Round 3

Reviewer 2 Report

Comments and Suggestions for Authors

Dear authors,

I really wanted to get answers to my questions. That's why I didn't find any answers about the yields of the obtained esters and looked at the contents of references 16 and 17.

It turned out that the reference [16] describes the acylation of microcrystalline cellulose, so the degrees of substitution cannot match ("The DS and yields are consistent with previously published articles"), since this is cellulose with a different degree of polymerization and other properties. Surprisingly, the molecular weight of microcrystalline cellulose from [16] completely matches "Fibrous cellulose" from the peer-reviewed article: two different names of cellulose with the same CAS No. 9004-34-6! Authors, you have a big problem. What are you going to edit now?

Please note that my question about the yield values remains UNANSWERED. But if the authors had answered, then the relationships between yields and degrees of substitution could have been checked, since the reviewer is sure that in the line "between 0.8 – 1.1 and CDA with … of 2.4" the values could be in the opposite order. The longer the substituent chain, the lower the degree of substitution.

Typos. Lines in the updated version 132-134. The given values "0.8 – 1.1 and … 2.4." refer to the degree of substitution (DS), not the degree of polymerization (DP), therefore there are typos in these lines.

Unfortunately, the authors answered two different questions 1 and 4 with the same text with typos.